# Planar chiral metasurfaces with maximal and tunable chiroptical response driven by bound states in the continuum

Tan Shi[1,7], Zi-Lan Deng[1,7 ✉], Guangzhou Geng[2,7], Xianzhi Zeng[1], Yixuan Zeng[3], Guangwei Hu[3], Adam Overvig[4], Junjie Li[2 ✉], Cheng-Wei Qiu[5], Andrea Alù[4], Yuri S. Kivshar[6] & Xiangping Li[1 ✉]

Optical metasurfaces with high quality factors (Q-factors) of chiral resonances can boost substantially light-matter interaction for various applications of chiral response in ultrathin, active, and nonlinear metadevices. However, current approaches lack the flexibility to enhance and tune the chirality and Q-factor simultaneously. Here, we suggest a design of chiral metasurface supporting bound state in the continuum (BIC) and demonstrate experimentally chiroptical responses with ultra-high Q-factors and near-perfect circular dichroism (CD = 0.93) at optical frequencies. We employ the symmetry-reduced meta-atoms with high birefringence supporting winding elliptical eigenstate polarizations with opposite helicity. It provides a convenient way for achieving the maximal planar chirality tuned by either breaking in-plane structure symmetry or changing illumination angle. Beyond linear CD, we also achieved strong near-field enhancement CD and near-unitary nonlinear CD in the same planar chiral metasurface design with circular eigen-polarization. Sharply resonant chirality realized in planar metasurfaces promises various practical applications including chiral lasers and chiral nonlinear filters.

[1] Guangdong Provincial Key Laboratory of Optical Fiber Sensing and Communications, Institute of Photonics Technology, Jinan University, 510632 Guangzhou, China. [2] Beijing National Laboratory for Condensed Matter Physics, Institute of Physics, Chinese Academy of Sciences, 100191 Beijing, China. [3] School of Electrical and Electronic Engineering, Nanyang Technological University, Singapore 639798, Singapore. [4] Photonics Initiative, Advanced Science Research Center, City University of New York, New York, NY 10031, USA. [5] Department of Electrical and Computer Engineering, National University of Singapore, Kent Ridge 117583, Republic of Singapore. [6] Nonlinear Physics Center, Research School of Physics, Australian National University, Canberra, ACT 2601, Australia. [7] These authors contributed equally: Tan Shi, Zi-Lan Deng, Guangzhou Geng. ✉email: zilandeng@jnu.edu.cn; jjli@iphy.ac.cn; xiangpingli@jnu.edu.cn

Chirality refers to the property of an object that cannot be superimposed with its mirror image after rotations or translations[1–3]. It is quite common in nature, e.g., in organic molecules[4], quartz crystals[5], and many others. The study of chirality is fundamentally important in various areas, including analytical chemistry, pharmaceutics, and even searching for extraterrestrial life. Light interactions with these geometries can induce chiroptical effects, including circular dichroism (CD) and optical activity, manifested by a difference in intensity and phase responses between left/right circularly polarized (LCP/RCP) light illuminations. Optical chirality was previously demonstrated in three-dimensional (3D) photonic structures[6], such as the helices[6,7], twisted cross structures[8,9] and multi-layered structures[10–12], all exhibiting strong CD and optical activity with preserved circular polarization, which however require very demanding 3D nanofabrication techniques. Recently, two-dimensional (2D) or planar structures have been shown to support both intrinsic planar chirality at normal incidence[13–16] and extrinsic chirality at oblique incidences[17–21]. Being different from 3D chirality with broken mirror symmetry in the propagation direction, although even existing in some stacked structures[15], planar chirality shows a circular polarization conversion between the output and incident light, and the cross-polarization behavior can be further exploited to modulate the geometric phase for arbitrary chiral wavefront shaping[22–26]. However, based on existing approaches it is challenging to achieve maximal planar chirality combined with ultra-high quality (Q-) factor due to absorption and scattering loss, hindering many applications that rely on strong chiral light-matter interactions.

Importantly, bound states in the continuum (BICs) can provide a feasible solution for significant problems of chiral photonics. BICs are identified as localized states coexisting with extended modes within the light cone, and they have attracted tremendous attention due to their unbounded Q-factors, which may boost light-matter interactions for applications such as surface-emitting lasing[27–29], biomedical sensors[30–32], and nonlinear frequency conventers[33–38]. BICs originate from destructive interference of several radiative channels, and hence they cannot be accessed externally. Quasi-BICs (q-BICs) with high-Q resonances are tailored perturbing the symmetry to couple out the resonant mode to free-space radiation[39–49]. Recently, chiroptical nanostructures mediated by BICs have been proposed, demonstrating perfect unitary chirality and ultrahigh Q-factors[50,51], and greatly expanding the available platforms to achieve optical chirality[51–57].

Here, we suggest and realize experimentally a design of planar chiral metasurfaces at optical frequencies with simultaneously mediated maximal chirality and ultrahigh Q-factors. A double-sided scythe (DSS-) shaped α-Si inclusion with in-plane inversion $C_2$ symmetry but without in-plane mirror symmetry is employed to construct the BIC state. Such BIC state with strong birefringence supports a vortex polarization singularity ($V$ point) surrounded by elliptical eigenstate polarizations with non-vanishing helicity, providing a convenient way to achieve maximum and high-Q chirality by slightly perturbing the inversion symmetry. With either breaking in-plane inversion symmetry or varying illumination symmetry, we realize planar chiral q-BIC states with strong intrinsic or extrinsic chirality, exceeding CD = 0.99 (in simulation) and CD = 0.93 (in experiment). For the intrinsic chirality, the Q-factor of the CD spectra has an inverse quadratic relation with respect to the geometrical asymmetry, with a nearly unchanged CD peak. For the extrinsic chirality, both value and sign of CD can be flexibly tuned by changing the incident angles encircling the Γ point, which is connected with the intrinsic non-vanishing helicities of the eigen-polarizations

near the BIC singularity[58]. It reveals that the maximal linear CD does not necessarily occur at the q-BIC with circular eigen-polarization due to the interference with background scattering channels; while the maximal nonlinear CD can be achieved exactly at the circularly polarized q-BIC with strong near-field enhancement CD based on the same planar chiral metasurface design. We believe that the demonstrated planar chiral metasurfaces governed by the BIC physics may find many applications in chiral lasers, nonlinear filters, and other active chiroptical devices.

## Results

Most previous q-BIC responses in metasurfaces usually did not support polarization effects[28,43,44] (Fig. 1a) or only supported linear polarization selectivity[30,39,41]. Very recently, 3D optical chiral q-BICs with versatile polarization responses (chirality selectivity) were theoretically proposed by breaking the out-of-plane symmetry with stereo-structures[50,51,57] (Fig. 1b). Our present work shows that planar metasurfaces with only reduced in-plane symmetry can support 2D extrinsic and intrinsic chirality with q-BICs (Fig. 1c). Figure 1d shows the geometry of interest, a double-sided scythe (DSS) α-Si structure (refractive indexes refer to Supplementary Fig. S2) employed as the unit cell of our proposed planar metasurface. Such DSS structure ($W_1 = W_2$ and $L_1 = L_2$, but $L_1 \neq W_1$) has in-plane inversion ($C_2$) symmetry, but lacks any in-plane mirror symmetries. From the simulated band structure and quality factor as shown in Fig. 1e, f, we see that the DSS metasurface hosts a BIC at the Γ point, characterized by a vertical magnetic dipole (MD) mode as shown in Fig. 1e, f. Different from the Γ point BIC state supported by highly symmetric inclusions[58], which is companied by a polarization singularity surrounded by vortex linear polarizations, the BIC state of the DSS structure supports a polarization singularity enclosed by a vortex elliptical eigen-polarization with non-vanishing helicity (see the inset of Fig. 1f and Supplementary Fig. S3). The planar chirality behavior can be predicted by temporal coupled mode theory (TCMT) (Supplementary Notes I & II), which suggests that a perfect planar chiral response can be obtained by tuning the local response to be birefringent using a geometry that has an inversion center, and then breaking inversion symmetry with a suitable perturbation to yield an intrinsic planar chiral q-BIC. Our proposed DSS inclusion with reduced symmetry provides a smart perturbation degree of freedom along the 45° direction, capable of achieving both maximal extrinsic and intrinsic chirality by perturbing a single parameter. As a general remark, in our systems the sign of the CD is determined by the sign of the helicity, while the maximum of CD does not necessarily coincide with circular eigen-polarization with the highest helicity, because there is non-zero background scattering accumulated by Jones matrix elements (see Supplementary Notes I and Supplementary Fig. S4).

Based on such a BIC state supported by DSS inclusion with reduced symmetry, one can expect both extrinsic and intrinsic planar chirality by slightly breaking the BIC symmetry. Figure 2a–d show the q-BIC with near-unity extrinsic chirality achieved by illumination symmetry breaking (varying incident angle θ and conical angle φ). Here, the CD is defined as the transmittance difference under right-handed polarization (RCP) and left-handed polarization (LCP) incidence [Eq. 1]:

$$CD = \frac{(T_{rr} + T_{lr}) - (T_{rl} + T_{ll})}{(T_{rr} + T_{lr}) + (T_{rl} + T_{ll})}, \quad (1)$$

where $T_{ij} = |t_{ij}|^2 (i = r, l; j = r, l; r$ represents RCP, $l$ represents LCP) is the transmittance of output polarization $i$ from the

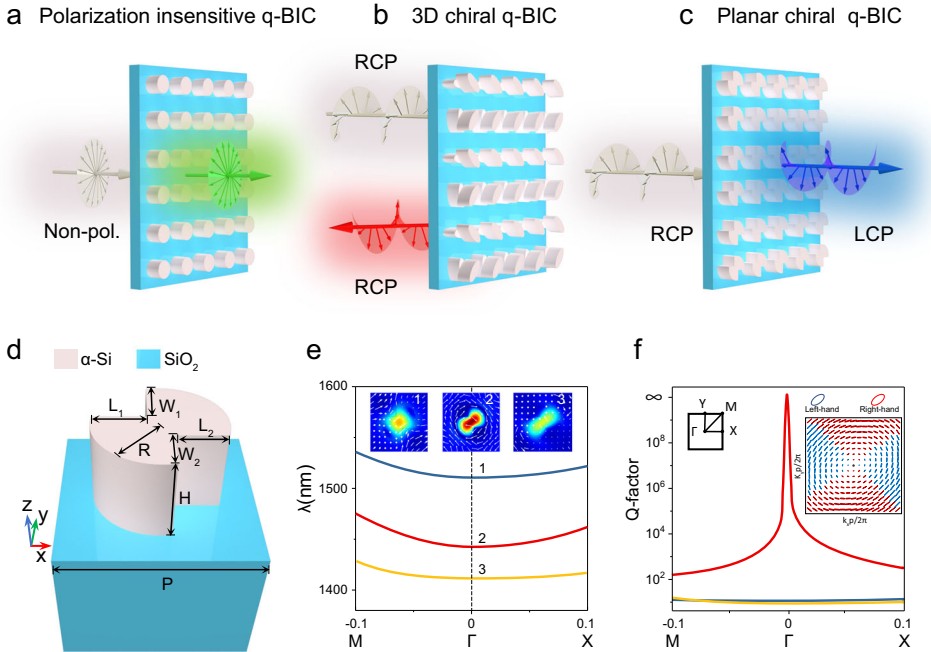

**Fig. 1 Schematics of different BIC states and planar chiral q-BIC metasurface with a polarization singularity enclosed by vortex elliptical eigen-polarizations with non-vanishing helicities. a–c** Comparison of q-BICs (**a**) in planar metasurfaces without polarization effect, (**b**) in stereo-nanostructures with 3D chiral effect in reflection, and (**c**) in planar metasurfaces with 2D chiral effect in transmission. **d** Schematic of the unit cell of the planar chiral q-BIC metasurface formed by the amorphous silicon (α-Si) DSS structure on a SiO$_2$ substrate with parameters $P = 850$ nm, $R = 280$ nm, $L_1 = L_2 = 220$ nm, $W_1 = W_2 = 191$ nm, and $H = 350$ nm. **e** Simulated band structure with real parts of the eigen-wavelength of the DSS metasurface near the Γ point. Insets show the field patterns (color: magnetic distributions, arrow: electric vectors) of the three typical modes. **f** Simulated Q-factors of the eigenmode in the k-space in the vicinity of the Γ point. Insets show the first Brillouin zone and eigen-polarization profiles. On the eigen-polarization profile, the polarization states are represented by ellipses of which the blue and red represent the eigen left-handed and right-handed states, respectively, and the black dot represents the V point (BIC).

input polarization $j$, and all element of $t_{ij}$ construct the Jones matrix under circular polarization basis [Eq. 2][11,59]

$$J_{circ} = \begin{pmatrix} t_{rr} & t_{rl} \\ t_{lr} & t_{ll} \end{pmatrix}. \qquad (2)$$

Figure 2b shows the calculated transmission spectra of all Jones matrix elements and the CD spectrum of the DSS metasurface with C$_2$ symmetry ($W_1 = W_2$ and $L_1 = L_2$, but $L_1 \neq W_1$) at oblique incidence ($\theta = 8°$, $\varphi = 90°$). $T_{lr}$ has a sharp peak >0.9, while all the other three Jones matrix elements ($T_{rl}$, $T_{ll}$ and $T_{rr}$) exhibit dips close to 0 at the resonant wavelength (1456 nm) of the q-BIC, resulting in an ultra-sharp CD spectrum with a maximum >0.95 (green curves in Fig. 2b). Figure 2c shows the evolution of CD spectra by continuously varying the incident angle $\theta$ along the $y$-direction ($\varphi = 90°$) (evolutions of transmission spectra of all Jones matrix elements are shown in Supplementary Fig. S5). The linewidth of the CD spectrum increases from 0 to finite values as $\theta$ increases, and the peak CD sustains near-unitary value with finite linewidths, manifesting the transfer process from a non-radiative BIC to a high-Q radiative q-BIC with strong extrinsic chirality. The Q-factor of the extrinsic chiral q-BIC shows the expected inverse quadratic law with the illumination asymmetry parameter $\alpha_1 = \sin\theta$ (Fig. 2d), which provides a convenient way to tailor the Q-factor of the chiral response. The evolutions of the extrinsic chiral q-BIC along the $x$-direction ($\varphi = 0°$) are shown in Supplementary Fig. S6, the variation trend is the same as the $\varphi = 90°$ case, except that the sign of CD is flipped due to the opposite helicity of eigen-polarization supported by the DSS structure along $x$- and $y$-directions (the inset of Fig. 1f). To investigate the full CD flip picture, the evolution of CD spectra

with varying conical angle $\varphi$ at a fixed incident angle $\theta = 12°$ is shown in Supplementary Fig. S7. The opposite varying trend of $T_{lr}$ and $T_{rl}$ can be observed as $\varphi$ is varying, leading to the sign flip of CD at positions near $(2n-1) \times 45°$, ($n = 1,2,3,4$), which is consistent with helicity sign flip shown in the inset of Fig. 1f.

Beyond the extrinsic chiral behavior, we can also access intrinsic high-Q chiral behavior under normal incidence. This can be achieved by introducing an in-plane geometrical asymmetry parameter ($\delta = W_2 - W_1$) as shown in Fig. 2e. As the DSS structure supports strong birefringence along 45°, an in-plane perturbation that breaks inversion symmetry will provide a degree of freedom to obtain planar chirality, as TCMT suggests in Supplementary Note II. Figure 2f depicts the transmission spectra of all Jones Matrix elements and the CD spectra of the metasurface with $\delta = 40$ nm ($W_1 = 191$ nm, $W_2 = 231$ nm). The ultra-sharp resonant peak of $T_{lr}$ and resonant dips of $T_{rl}$, $T_{ll}$ and $T_{rr}$ are obtained at the wavelength of 1392 nm, resulting in a CD peak >0.99 and an ultra-narrow linewidth of 1.45 nm. We also confirm the planar chiral q-BIC modes by utilizing multipole expansions and studying the nearfield electromagnetic patterns of the modes (Supplementary Fig. S8), which further confirm that the high-Q planar chiral q-BIC states are dominated by the MD mode. Figure 2g, h show the CD evolution spectra as a function of the asymmetry parameter $\delta$, the Q-factor of the CD spectrum also manifests an inverse quadratic relationship against the relative asymmetry parameter defined by $\alpha_2 = \delta/W_1$, which is the signature of symmetry-protected BICs (corresponding spectra for all components of the Jones matrix spectra refer to Supplementary Fig. S9). In addition, the circular eigen-polarization state gradually moves away from the Γ point when the asymmetry parameter $\delta$ is increasing, as shown in Supplementary Fig. S10. Different

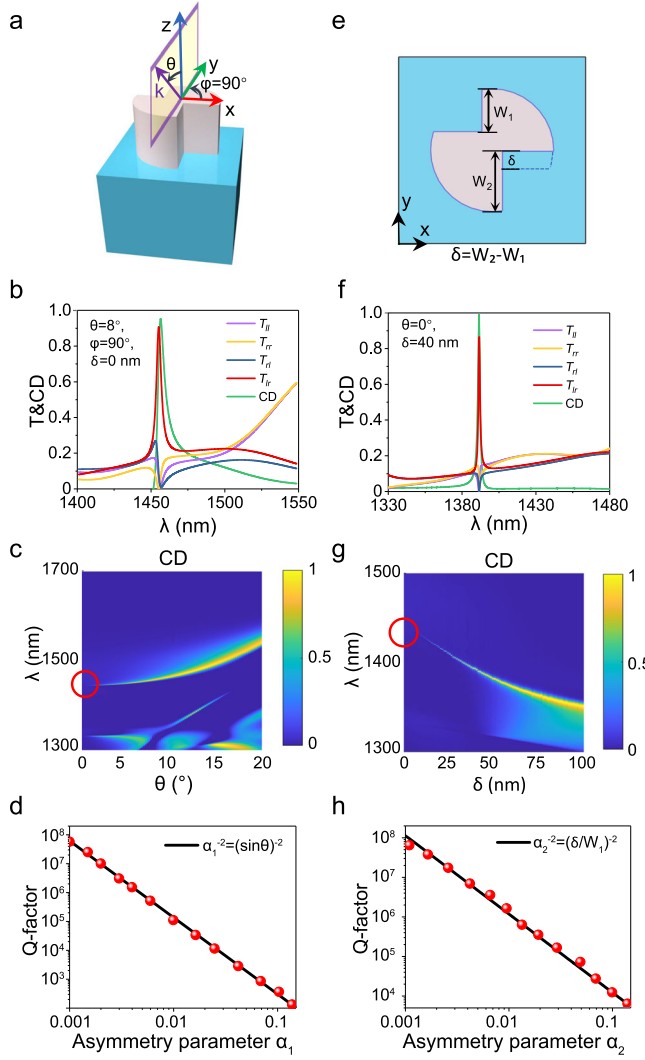

**Fig. 2 Proposed implementation of the DSS structure with both the extrinsic and intrinsic planar chirality by transferring the BIC to q-BICs.**
**a, e** Schematic of the symmetry-breaking processes that transfer the BIC to planar chiral q-BICs: (**a**) illumination symmetry breaking by varying incident angle $\theta$, (**e**) in-plane geometry symmetry breaking by $\delta = W_2 - W_1$.
**b, f** Simulated transmission Jones matrix spectra of $T_{ll}$, $T_{rr}$, $T_{rl}$ and $T_{lr}$ as well as the CD spectrum of the metasurface with (**b**) the same parameters as Fig. 1d ($\delta = 0$ nm) under oblique incidence ($\theta = 8°$, $\varphi = 90°$) and with (**f**) an asymmetric structure parameter ($\delta = 40$ nm) for normal incidences.
**c, g** The evolution of CD spectra by continuous varying (**c**) incident angle $\theta$ along the $\varphi = 90°$ direction and by (**g**) the geometry asymmetry parameter $\delta$. **d, h** Dependence of the Q-factors of the planar chiral q-BIC mode on the relative asymmetry parameter (**d**) $\alpha_1 = \sin\theta$, (**h**) $\alpha_2 = \delta/W_1$ around the chiral q-BIC state. The solid line shows an inverse quadratic fitting.

from the intrinsic 3D chirality with preserved helicity between input and output polarizations, the intrinsic planar chirality is always accompanied by circular conversion dichroism (CCD) and asymmetry transmission (AT)[2,13,60], as shown in Supplementary Fig. S11, due to the preserved symmetry in the propagation direction. The CCD is produced by the mutual orientation of chiral elements and light transmission direction, and the peak value of CCD is close to 1, which is slightly larger than CD (Supplementary Fig. S12). Perfect unitary circular conversion transmittance $T_{lr}$ and CD can be achieved with a strictly symmetric background as shown in Supplementary Fig. S13. The existence of a practical SiO$_2$ substrate introduces a perturbation of

the background scattering, resulting in reduced transmittance and CD, as presented in Fig. 2b, f.

Due to the symmetry-protected origin of this phenomenon, our designed DSS metasurface has good tolerance against fabrication imperfections, as tested in Supplementary Fig. S14, which provides a convenient way for experimental realization. We fabricated the DSS metasurfaces with different in-plane asymmetry parameters ($\delta = 0$, 20, 40, 60, and 80 nm), and employed a homemade optical setup (Fig. 3a) to measure the Jones matrix spectra of the metasurface. We first examine the extrinsic chiral q-BIC by measuring the DSS metasurface without in-plane geometry asymmetry ($\delta = 0$) for varying incident angles (Fig. 3b). Figure 3c shows the comparison between simulated and measured transmission Jones matrix spectra at different incident angles ($\theta = 2°$, 4°, 6°, 8°, 10°, 12°) along the $\varphi = 90°$ direction. At normal incidence (middle panel of Fig. 3c), all transmission spectra are smooth without resonant features, because the BIC state at the $\Gamma$ point is decoupled from any scattering channels of the external environment. At other incident angles, sharp circular conversion peaks of $T_{lr}$ and resonant dips of other three Jones matrix elements ($T_{ll}$, $T_{rl}$, $T_{rr}$) emerge in both simulations and experiments. The measured absolute peak transmittances are >0.78 for all incident angles, which is a little below the simulated 0.9, mainly due to the practical scattering loss of the fabricated sample as well as a limited collection efficiency of the measurement setup. As expected, we can clearly observe the linewidth increment of the q-BIC peak with increasing incident angle in the experiment, consistent with our theory and simulations. From the measured Jones matrix spectra, we extracted the CD spectra for different incident angles corresponding to Eq. (1) in Fig. 3d. As we can see, sharp CD peaks appear at the q-BIC resonant wavelengths, the simulated/measured CD maxima reach 0.67/0.61, 0.80/0.72, 0.95/0.83, 0.95/0.82, 0.96/0.86 and 0.96/0.88 with the simulated/measured Q-factors of 1835/602, 616/258, 261/142, 143/96, 95/56 and 58/39 for incident angles $\theta = 2°$, 4°, 6°, 6°, 10°, 12°, respectively. For normal incidence, the CD is always 0 as the BIC state is not accessed by external excitation. We also experimentally characterized the extrinsic chiral q-BIC along the $\varphi = 0°$ direction in Fig. S15, where sharp transmission peaks of $T_{rl}$ are observed, achieving opposite signs of simulated/measured CD: $-0.89/-0.82$, $-0.95/-0.88$, and $-0.97/-0.91$ with the simulated/measured Q-factors of 798/312, 196/102, 85/55 at incident angles $\theta = \pm4°$, $\pm8°$, $\pm12°$, respectively, which further confirms the CD flip behavior.

To verify the intrinsic chiral q-BIC further experimentally, we characterized the DSS metasurfaces with different in-plane geometrical asymmetries, as shown in Fig. 4. Figure 4a shows the scanning electron microscope (SEM) images of fabricated samples with different asymmetry parameters $\delta$. From both the top view and slant view, we see that the fabricated DSS silicon pillars are smooth and uniform, in agreement with the theoretical design. Figure 4b, c show the simulated and measured transmission Jones matrix and CD spectra of the five fabricated metasurfaces. When $\delta = 0$ (lowest panel in Fig. 4b, c), there is no resonance with $T_{rl} = T_{lr}$ and $T_{ll} = T_{rr}$ due to reciprocity, and CD = 0 crossing the whole frequency band. As the asymmetry parameter $\delta$ becomes non-zero, sharp circular conversion transmission peaks of $T_{lr}$ and CD appear, and the bandwidth increases with the increase of the asymmetry parameter (upper four panels in Fig. 4b, c). The simulated/measured CD maxima for the intrinsic planar chirality are 0.98/0.88, 0.99/0.93, 0.99/0.92, and 0.99/0.91 with the simulated/measured Q-factors of 2457/390, 480/121, 198/62, and 128/38 for $\delta = 20$, 40, 60, and 80 nm, respectively. We note that, the Q-factor can reach the order of $10^6$ with still relatively high CD of 0.88 by further decreasing the asymmetry parameter to $\delta = 2$ nm, as the simulation results in Supplementary Fig. S16 indicate. Such simultaneously achieved

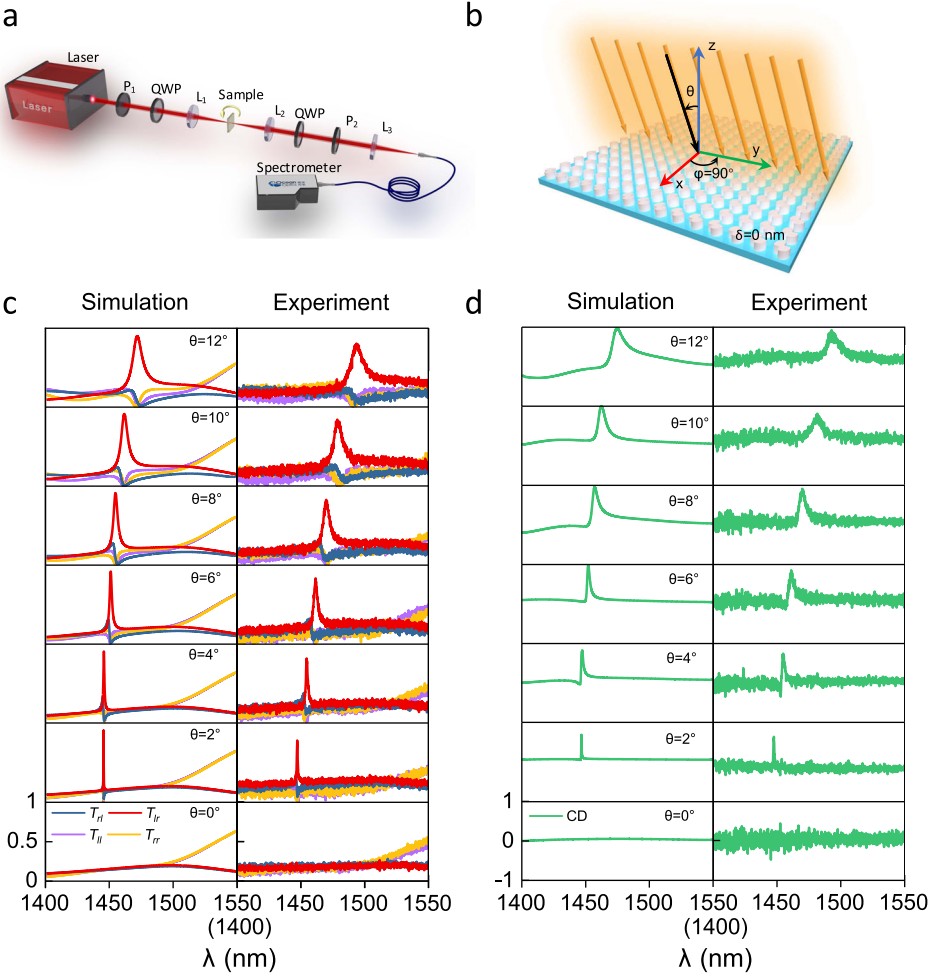

**Fig. 3 Experimental verification of the extrinsic planar chiral q-BIC with illumination symmetry breaking. a** Experimental setup for the Jones matrix spectra measurement under the circular polarization basis. $P_1$ and $P_2$ represent the polarizers, QWP is the quarter-wave plate, $L_1$, $L_2$, and $L_3$ are lenses. **b** Schematic of the metasurface with $\delta = 0$ supporting chiral q-BICs through different incident angles along the $\varphi = 90°$ direction. **c** Simulated and measured transmission Jones matrix spectra ($T_{lr}$, $T_{rl}$, $T_{rr}$ and $T_{ll}$) for different incident angles ($\theta = 0°$, 2°, 4°, 6°, 8°, 10°, 12°) along the $\varphi = 90°$ direction. **d** Simulated and measured CD spectra for different incident angles extracted from the Jones matrix spectra in (**c**).

strong chirality and high Q-factors promise chiroptical functionalities with strong light-matter interactions.

The planar chiral q-BIC metasurface can realize not only linear CD maximum, but also maximal nonlinear CD with promising active chiral applications. To pursue the active chiral behavior, a circularly polarized eigenstate whose excitation is independent of incident polarization is highly required. Based on the DSS-shaped design, we have found another set of parameters ($P = 780$ nm, $R = 268$ nm, $L_1 = 197$ nm, $L_2 = 200$ nm, $W_1 = 217$ nm, $W_2 = 257$ nm, and $H = 654$ nm) for generating the circular eigen-polarization state (Fig. 5). Figure 5a shows its transmission spectra under LCP/RCP normal incidences. The sharp resonance peak only exists for RCP incidence, while the overall spectrum is smooth for LCP incidence. It means that the q-BIC resonances could only be selectively excited by the RCP light, indicating a RCP eigenstate. Figure 5b showcases that the eigen-polarization at the Γ point is indeed a circular eigen-polarization (*C* point). Although the linear CD is not high (0.6) at the C point, the near-field enhancement contrast between RCP/RCP incidences is very large (400:1) as shown in Fig. 5c. Such huge near-field enhancement CD is the key to achieving active chiral applications such as chiral lasing, non-linear chiral emissions based on strong chiral light-matter interaction. We also confirm the planar chiral q-BIC modes by utilizing multipole expansions and study the near-field electromagnetic

patterns of the modes for LCP and RCP incidences in Supplementary Fig. S17. Different from the largest linear CD caused by MD mode, the largest nonlinear CD is dominated by ED mode, which is selectively excited by the RCP incidence. To confirm the active chiral emission behavior, we examined the nonlinear CD in both simulation and experiment. Figure 5d shows the simulated results of third harmonic generation (THG). The THG intensity under RCP pumping is significantly enhanced at the q-BIC resonance frequency. While the THG signal under LCP pumping is negligible. Therefore, the extracted nonlinear CD reaches unity at the resonance frequency. Figure 5e, f show the experimental results for both linear CD and nonlinear CD. Figure 5e shows the measured transmission spectra and linear CD. The transmission spectra for LCP incidence are not ideally smooth, which can be attributed to the imperfect circular eigen-polarization caused by fabrication deviations. Figure 5f shows the experimentally measured THG intensity under RCP/LCP pumping. Clear THG emission contrast yields nonlinear CD as high as 0.81. The THG efficiency is significantly enhanced by the q-BIC, as compared to the THG emission intensity by a reference silicon thin film. Limited by the operating band and loss of the objective lens, the effect of THG visible to the naked eye can be realized when the incident power is large enough, as shown in the inset of Fig. 5f. Figure 5g illustrates the log-log plot of emission power as a function of

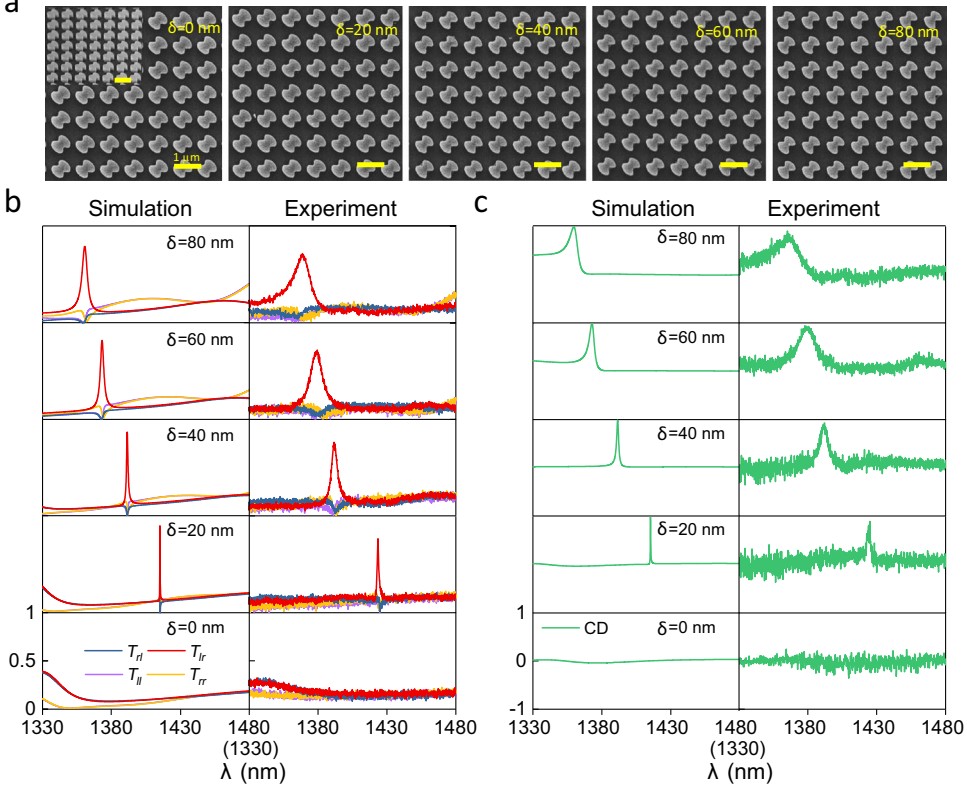

**Fig. 4 Experimental verification of the intrinsic planar chiral q-BIC with various in-plane geometry asymmetry parameters. a** The top view of the SEM image of five fabricated DSS metasurfaces with asymmetry parameters $\delta = 0$ nm, 20 nm, 40 nm, 60 nm, and 80 nm, respectively. The inset shows the slant view of the metasurface. Scale bar: 1 μm. **b** Simulated and measured transmission Jones matrix spectra ($T_{lr}$, $T_{rl}$, $T_{rr}$ and $T_{ll}$) of the five DSS metasurface samples with different $\delta$. **c** Simulated and measured CD spectra for different $\delta$ extracted from the Jones matrix spectra in (**b**).

pumping power showing a cubic power dependence, which is the signature of THG. TH conversion efficiency can be significantly enhanced by decreasing the asymmetry parameter of the chiral q-BIC as shown in Supplementary Fig. S18, which is attributed to the tunable and ultra-high Q-factor provided by the q-BIC. Therefore, our DSS structure design can satisfy not only the perfect linear chirality but also the nonlinear chirality. The maximal linear CD is not necessarily related to the circular eigen-polarization state due to the existence of background scattering. However, the non-linear CD and active chiral behaviors heavily rely on the circular eigen-polarization, which could indeed be achieved by the planar chiral q-BIC metasurface, not necessarily relying on 3D chiral structures.

## Discussion

We have suggested and realized experimentally planar chiral metasurfaces with in-plane arrays of symmetry-reduced meta-atoms. Such metasurfaces are governed by the physics of bound states in the continuum with strong birefringence, identified as a vortex polarization singularity surrounded by ellipse eigenstate polarizations with non-vanishing helicity. Both extrinsic and intrinsic planar chirality can be realized by symmetry breaking. By illumination symmetry breaking, the BIC state can be trans-ferred to q-BIC with strong extrinsic chirality accompanied by tunable linewidth and signs. By introducing in-plane structure asymmetry, the intrinsic planar chirality can be achieved under normal incidence with maximum CD of 0.99 (in theory) and 0.93 (in experiment) or ultra-high nonlinear CD of 0.99 (in theory) and 0.81 (in experiment) at optical frequencies. Beyond the chirality based on circular polarization, we can envision any elliptical chirality by varying simultaneously birefringence and

axis angle of the meta-atoms. Due to the high Q-factor of the planar chiral q-BIC in metasurfaces with its accessibility and controllability, our results provide unique opportunities for many applications requiring ultrahigh Q-factors and chirality control, including chiral lasing and chiral nonlinear optics.

## Methods

**Simulation of reflection and transmission efficiency.** Rigorous coupled wave analysis (S4)[61] simulations were carried out to simulate the planar chiral q-BIC metasurface and to calculate efficiencies for the co-polarization handedness-pre-served coefficients, and the circular cross-polarization conversion coefficients transmission spectra. The metasurfaces are designed by α-Si columns based on a $SiO_2$ substrate.

**Fabrication of samples.** The metasurfaces were fabricated on a fused quartz substrate by utilizing electron beam lithography and including the processes such as deposition, patterning, lift-off, and etching. First, a 350 nm-thick amorphous silicon (α-Si) film was deposited by plasma-enhanced chemical vapor deposition method at 120 °C, during which the flow rates of $SiH_4$ and Ar are 10 and 475 sccm. And the deposition pressure and RF power are 650 mTorr and 20 W, respectively. Then a PMMA film of 300 nm was spinning coated and covered by PEDOT: PSS film as a conducting layer. The desired structure was patterned by utilizing JEOL 6300FS EBL at a base dose of 1000 μC/cm[2] with an accelerating voltage of 100 kV for 1 h. After the exposure process, the conducting layer was washed away and the resist was developed in 1:3 MIBK: IPA solution for 40 s and rinsed in IPA for 30 s successively, followed by a deposition of 80 nm Cr using electron beam evaporation deposition method. To realize the lift-off process, the sample was immersed in hot acetone of 75 °C and cleaned by ultrasonic. Finally, by using inductively coupled plasma (ICP) reactive ion etching (RIE) method with HBr at room temperature (RT) for 260 s (flow rate of 50 sccm, the pressure of 10 mTorr, RF and ICP power of 50 and 750 W, respectively), the desired structure was transferred from Cr to silicon and the residual Cr was removed by cerium (IV) ammonium nitrate. The five planar chiral q-BIC metasurfaces are composed of 500 × 500 periods (425 μm × 425 μm) with different cross-sections ($\delta = 0$ nm, 20 nm, 40 nm, 60 nm, and 80 nm), and the final structures are shown in Fig. 4a.

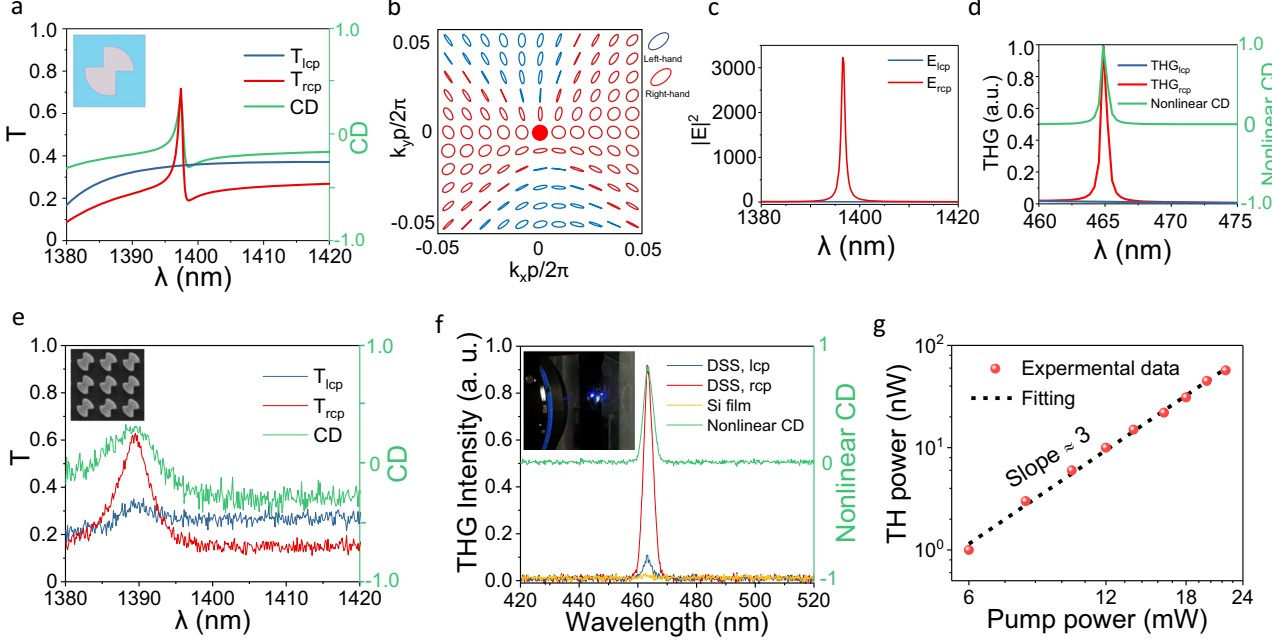

**Fig. 5 Near-unity nonlinear CD with planar chiral q-BIC metasurface. a** Simulated transmission spectra of $T_{lcp}$ and $T_{rcp}$ as well as the linear CD spectra of the metasurface with parameters $P = 780$ nm, $R = 268$ nm, $L_1 = 197$ nm, $L_2 = 200$ nm, $W_1 = 217$ nm, $W_2 = 257$ nm, and $H = 654$ nm with an asymmetric structure parameter ($\delta = 40$ nm) for normal incidences. Insets show the Structure diagram. **b** The eigen-polarization profile in the $k$-space. The red dot represents the circular eigen-polarization ($C$ point). **c** Contrast of electric field enhancement $|\mathbf{E}|^2$ of different circularly polarized light. **d** Simulated THG intensity under different circularly polarized light incidence and the corresponding nonlinear CD. The nonlinear CD is defined as the normalized THG difference between RCP and LCP lights. **e** Measured transmission spectra of $T_{lcp}$ and $T_{rcp}$ as well as the linear CD spectrum. Insets show the SEM image of the metasurface sample. **f** Experimentally measured THG intensities under RCP, LCP incidences in planar chiral q-BIC metasurface, THG intensity in a reference Silicon thin film, as well as the measured nonlinear CD spectra. The inset shows a photographic image of the light spot of THG from the sample. **g** Power dependence of THG in logarithmic scale showing cubic power scaling law. The red dots show the measured data, the black dashed line is a fit to the data with a third-order power function.

**Characterization of samples**. A supercontinuum laser (Fianium-WL-SC480) was employed as the broadband light source for the measurement of the transmission spectra of all polarization components under circular polarization incidences. Incident light with a circular polarization was generated by cascading a broadband polarizer and a quarter waveplate from the supercontinuum laser. Then, the incident light was focused on the sample by a lens with a focal length of 5 cm. Subsequently, the polarization components are detected by a quarter waveplate and a polarizer. The transmittances were measured by means of an Ocean spectrometer (flame-NIR). Moreover, the sample was mounted on a rotation stage for varying incident angles in the measurement. In the nonlinear optical measurements, we use a spectrally tunable fs laser source (repetition frequency: 80 MHz, pulse duration: ~146 fs). Incident light with the circular polarization was generated by cascading a broadband polarizer and a quarter waveplate. The laser beam was focused on the sample surface by an objective with effective NA of about 0.42. The transmitted light was collected on the other side by using another objective with NA = 0.95 and then was sent to an Ocean spectrometer (USB4000) for spectrum analysis.

## Data availability

The data that support the findings of this study are available from the corresponding author upon reasonable request.

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

## Acknowledgements

The authors acknowledge the funding support provided by National Key R&D Program of China (2021YFB2802003), National Natural Science Foundation of China (NSFC) (62075084, 12074420), Guangdong Basic and Applied Basic Research Foundation (2022B1515020004, 2020A1515010615), Fundamental Research Funds for the Central Universities (21620415), Guangzhou Science and Technology Program (202102020566), Guangdong Provincial Innovation and Entrepreneurship Project (2016ZT06D081), Beijing Municipal Science&Technology Commission &Administrative Commission of Zhongguancun Science Park (Z211100004821009), Air Force Office of Scientific Research, Vannevar Bush Faculty Fellowship, Simons Foundation, and Australian Research Council (grant DP210101292).

## Author contributions

Z.-L.D. and T.S. conceived the idea. T.S., Z.-L.D., and X.L. designed the experiments. T.S., Z.-L.D. carried out the design and simulation of the metasurface. T.S., Z.-L.D., G.H., A.O., and Y.Z. conducted the theoretical analysis of the results. G.G. and J.L. fabricated the samples. T.S. and X.Z. performed the measurements. Z.-L.D., Y.K., A.A., and X.L. supervised the project. Z.-L.D., T.S., and X.L. analyzed the data and wrote the paper. All authors contributed to the revision and discussion of the paper.

## Competing interests

The authors declare no competing interests.
