## [Peer Review File · Nature Communications]

REVIEWER COMMENTS

Reviewer #1 (Remarks to the Author):

The manuscript by Shi et. al. describes the design, fabrication and characterization of a new metasurface structure exhibiting a quasi-bound-state-in-continuum (qBIC) resonance. qBICs have become very popular within the metasurface community over the last few years thanks to their ability to generate tunably high Q factors. That is to say, the Q factor can be mapped to some structural or configurational parameter allowing arbitrarily high Q to be realized in principle. As the authors mention, the circulating light intensity within a resonator scales with Q, so high Q structures are useful for achieving strong light-matter interaction strengths. The novelty of the current manuscript is that the authors experimentally demonstrate a qBIC metasurface with a high Q resonance that discriminates between different circular polarizations, measured as narrowband very strong circular dichroism. This is realized by starting with a C2 symmetric metasurface supporting a BIC and then subtly breaking the inversion symmetry to simultaneously tune Q and introduce chirality.

The paper is well written, the physics clearly explained, and the data nicely presented. I also believe that the authors' claim of having achieved the record Q factor for a circularly dichroic structure is valid. However, from a theoretical/conceptual perspective, I'm not convinced that the current structure is that novel. Other numerical papers have appeared, including those from the authors themselves, showing qBIC based circular dichroism using very similar symmetry breaking arguments as applied here. I find the claim that previously proposed designs were insufficient because they are based on "complicated stereo-nanostructures, making elusive an experimental realization and application development, especially at optical frequencies" strange. Many metasurfaces with a similar Q to here have been realized with stereo-nanostructures, and I see no reason why the present structure would be more robust than a planar stereo-structure. Because of this, I believe the importance of this paper is defined by the practical significance of high Q circular dichroism.

The authors state that their result has "many applications in chiral sensing of biomolecules, spin selective emitters, and active chiroptical devices". However, it is important to note that all measurements of chirality in the paper are of far-field quantities. Chiral sensing of biomolecules and enhancing emission from spin selective emitters requires circularly polarized nearfields, which has not been demonstrated in the paper for this structure. Likewise, for lasing, the eigenmodes of the resonator are important, not just the Jones matrix amplitudes. Combining polarization control with high Q nonlinearity enhancement may be fruitful, but the dichroism observed is also specifically between the conversion from left-handed to right-handed CPL vs right-handed to left-handed. It is not immediately obvious to me why efficient nonlinear circular conversion dichroism would be useful.

In summary, while I find the result interesting, to meet the bar for Nature Communications a much more careful and convincing discussion should be added explaining specifically how the high Q circular conversion dichroism observed will be useful or lead to new directions.

For completeness I've included a few specific comments below:

Caption to figure 1 "tree typical modes" should presumably be "three typical modes".

This passage is misleading: "Although very recently q-BICs with versatile polarization responses (chirality selectivity) were theoretically proposed, they required complicated 3D structures with broken symmetry in the propagation direction^{50,51,58} (Fig. 1b), hindering their implementation within current 2D patterning technology, especially at optical frequencies. Our present work shows that planar metasurfaces with only reduced in-plane symmetry can support q-BICs with both extrinsic and intrinsic planar chirality". It implies that the new structure gives access to the same chiral optical response as these 3D structures just with simpler fabrication. This isn't true and should be clarified.

It would be useful to check and include angles and values of the parameter delta smaller than 4 degrees and 20 nm to explicitly show the limits of Q for these structures.

I think it would be informative to plot the polarization eigenstate spectra for the simulated data for different parametric distances from the BIC. (This may also illuminate possible applications)

Reviewer #2 (Remarks to the Author):

The authors demonstrate silicon metasurfaces sustaining high Q-factor modes that exhibit extrinsic and intrinsic chiral response. As the structure is planar, the CD is accompanied by circular polarization conversion and asymmetric transmission. The Q-factor of the metasurface is on the order of 10^2 . The results are technically sound, obtained with appropriate techniques, analyzed and interpreted very carefully, and presented in excellent detail. The conclusions are well supported by the results. However, chiral planar metasurfaces have been demonstrated in the literature (as also acknowledged by the

authors e.g. refs. 14,15,24,38). Some of these featured high-Q resonances (e.g. Fano resonance in ref.14, q-BIC in ref.38). For these reasons I cannot recommend the publication of the submitted article in its present form to Nature Communications as it lacks of significant advancements compared to the established literature.

A couple of additional points:

1. From the simulation results in Fig. 1e the BIC show a strong dispersion as the in-plane wave vector varies. What is the NA used in the measured transmittance spectra? Have the authors theoretically considered how a finite NA would affect the optical response?
2. The result in the abstract do not seem to match the main results of the paper. For the metasurface with Q-factor 390 it reads CD equal to 0.88.

Reviewer #3 (Remarks to the Author):

In this work, by means of numerical simulations the authors design a dielectric metasurface that exhibits a narrow resonance with high chirality. Also, the proposed system is experimentally verified, showing quantitatively and qualitatively agreement between simulations and measurements.

The design is based on the concept of bound states in the continuum (BICs) and asymmetric meta-atoms (without in-plane mirror symmetry). First, the eigenfrequencies of the proposed system is studied, showing a resonance with divergence Q-factor associated with the well-known symmetry-protected BIC at the Gamma point (in this system, related to the magnetic dipole response). Later, the circular dichroism (CD) is calculated using two different mechanisms to couple to the quasi-BIC are analyzed: (i) by changing the angle of incidence; (ii) by breaking the in-plane inversion symmetry of the meta-atoms (and working at normal incidence). In both cases narrow resonances with almost "CD = 1" are found in the proximities of the BIC condition. Finally, the theoretical design is experimentally tested, validating the numerical results.

I personally think that the manuscript is solid, rigorous, well written and well presented. The good agreement between simulations and experiment makes the work appealing. In addition, the results contribute to the progress in the current field. For these reasons I believe that the manuscript deserves publication in Nature Communications. Nonetheless, I have some minor concerns that would help to polish the manuscript.

1) Ref. 58 is included in the sentence

- “Although very recently q-BICs with versatile polarization responses (chirality selectivity) were theoretically proposed, they required complicated 3D structures with broken symmetry in the propagation direction”.

In my understanding, in this work the proposed structure is not a “complicated 3D structure”, it is simply made of rods with slightly different heights and it can be considered a planar chiral metasurface. In order to improve the novelty statements of the current manuscript, I suggest introducing this reference in a different way.

2) In the first section of the manuscript all the results are obtained by means of numerical simulations, but this fact is not clearly said in the text. I encourage the author to include a sentence highlighting this point.

3) Again, in the conclusions it is said that

- “the intrinsic planar chirality can be achieved under normal incidence with maximum CD of 0.99 (theory) and 0.93 (experiment)”.

It should be read “simulation” instead of “theory”.

4) From the eigenmode analysis, the metasurface supports three resonances (at the frequency windows under study), but only the BIC resonance is observed in the CD maps of Fig. 2 c and g. Could the authors comment on this fact? I would also expect some (broad) features around the other resonant frequencies since the meta-atom is chiral per se.

5) It looks like the experimental results shown in Fig.3 and 4 are noisier for low transmittance than for higher one. Then, the narrow resonances are well resolved, but one can think that noise is “numerically” removed . It would be valuable to include some comments about that.

6) It is interesting that in the comparison of simulations and experiments, the results of Fig. 3 (illumination symmetry breaking) shown similar resonance widths, while for Fig. 4 (broken meta-atom

in-plane inversion symmetry) the experiment is broader than simulation. Can this be related to the structure tolerance as studied in Fig. S13?

7) In the "Author Contribution" section, "G. H. A. O" is written (without the comma that separates the names).

Response to Reviewer #1

GENERAL COMMENTS

The manuscript by Shi et. al. describes the design, fabrication and characterization of a new metasurface structure exhibiting a quasi-bound-state-in-continuum (qBIC) resonance. q-BICs have become very popular within the metasurface community over the last few years thanks to their ability to generate tunably high Q factors. That is to say, the Q factor can be mapped to some structural or configurational parameter allowing arbitrarily high Q to be realized in principle. As the authors mention, the circulating light intensity within a resonator scales with Q, so high Q structures are useful for achieving strong light-matter interaction strengths. The novelty of the current manuscript is that the authors experimentally demonstrate a qBIC metasurface with a high Q resonance that discriminates between different circular polarizations, measured as narrowband very strong circular dichroism. This is realized by starting with a C_2 symmetric metasurface supporting a BIC and then subtly breaking the inversion symmetry to simultaneously tune Q and introduce chirality.

The paper is well written, the physics clearly explained, and the data nicely presented. I also believe that the authors' claim of having achieved the record Q factor for a circularly dichroic structure is valid. However, from a theoretical/conceptual perspective, I'm not convinced that the current structure is that novel. Other numerical papers have appeared, including those from the authors themselves, showing qBIC based circular dichroism using very similar symmetry breaking arguments as applied here. I find the claim that previously proposed designs were insufficient because they are based on "complicated stereo-nanostructures, making elusive an experimental realization and application development, especially at optical frequencies" strange. Many metasurfaces with a similar Q to here have been realized with stereo-nanostructures, and I see no reason why the present structure would be more robust than a planar stereo-structure. Because of this, I believe the importance of this paper is defined by the practical significance of high Q circular dichroism.

The authors state that their result has "many applications in chiral sensing of

biomolecules, spin selective emitters, and active chiroptical devices”. However, it is important to note that all measurements of chirality in the paper are of far-field quantities. Chiral sensing of biomolecules and enhancing emission from spin selective emitters requires circularly polarized nearfields, which has not been demonstrated in the paper for this structure. Likewise, for lasing, the eigenmodes of the resonator are important, not just the Jones matrix amplitudes. Combining polarization control with high Q nonlinearity enhancement may be fruitful, but the dichroism observed is also specifically between the conversion from left-handed to right-handed CPL vs right-handed to left-handed. It is not immediately obvious to me why efficient nonlinear circular conversion dichroism would be useful.

In summary, while I find the result interesting, to meet the bar for Nature Communications a much more careful and convincing discussion should be added explaining specifically how the high Q circular conversion dichroism observed will be useful or lead to new directions.

Response: We thank Reviewer #1 for endorsing that our work is well written, the physics clearly explained, and the data nicely presented, the result is interesting to meet the bar for Nature Communications conditionally. To clarify the advantages of the proposed metasurface and how it could be useful or lead to new directions, we have supplemented it with theoretical analysis and new experimental data to strengthen our work.

Advantages: The principle behind the q-BIC is based on symmetry-breaking operation, which is in line with the literature. In this context, our current planar design does not necessarily out-perform the previous 3D structure and planar stereo-structure designs. Followed by the reviewer’s suggestion, we no longer emphasize that our current planar design out-performs the previous 3D structure and planar stereo-structure designs. The introduction of chirality into q-BIC offers great flexibility to extremely tune CD and Q-factor simultaneously, representing the unique advantage of the current work. Compared with 3D chiral stereo-nanostructures to break symmetry in the propagation direction, the proposed DSS planar structure allows the first experimental demonstration of high-Q and near-unitary CD with tunability by illumination angle or

structure asymmetry at optical frequencies.

New phenomena: In addition, further analysis reveals that our proposed planar chiral q-BIC metasurfaces are capable to achieve both linear and nonlinear maximal CD under the same planar symmetry-breaking operation. The former one does not necessarily rely on a circular eigen-polarizations due to interplay with background scattering in the far-field, while the later one occurs exactly on the circular eigen-polarizations with strong near-field enhancement discrimination under LCP/RCP excitations. With respect to the high-Q near-unitary linear CD, the flexibly tunable CD against illumination angle in the full range from -1 to 1 , and the regularly tailored Q-factor against asymmetry parameter provide an excellent platform for applications like far-field chiral scattering devices and frequency-selective asymmetric transmission devices. With respect to high-Q near-unitary nonlinear CD (Fig. R1), the associated chiral near-field enhancement and circular eigen-polarization may pave a new pathway for active chiral devices including chiral emitters and even chiral lasers. Both of them were experimentally achieved in the planar chiral metasurfaces in form of high-Q circular conversion dichroism. Therefore, no matter it is the 3D chiral structure with circular dichroism with preserved polarization response between the output and input light or the 2D chiral structure with circular conversion dichroism, one can obtain both linear CD in the far-field scattering and nonlinear CD based on near-field enhancement discrimination under different circular polarized illuminations. In this way, we could directly see efficient nonlinear circular conversion dichroism would be useful in our current planar chiral q-BIC metasurface designs. With these, we envision various potential applications in chiral sensing of biomolecules, spin selective emitters, and active chiroptical devices.

Fig. R1. Near-unity nonlinear CD with planar chiral q-BIC metasurface. (a) Simulated transmission spectra of T_{lcp} and T_{rcp} as well as the linear CD spectra of the metasurface with parameters $P=780$ nm, $R=268$ nm, $L_1=197$ nm, $L_2=200$ nm, $W_1=217$ nm, $W_2=257$ nm, and $H=654$ nm with an asymmetric structure parameter ($\delta = 40$ nm) for normal incidences. Insets show the Structure diagram. (b) The eigen-polarization profile in the k -space. The red dot represents the circular eigen-polarization (C point). (c) Contrast of electric field enhancement $|E|^2$ of different circularly polarized light. (d) Simulated THG intensity under different circularly polarized light incidence and the corresponding nonlinear CD. The nonlinear CD is defined as the normalized THG difference between RCP and LCP lights. (e) Measured transmission spectra of T_{lcp} and T_{rcp} as well as the linear CD spectrum. Insets show the SEM image of the metasurface sample. (f) Experimentally measured THG intensities under RCP, LCP incidences in planar chiral q-BIC metasurface, THG intensity in a reference Silicon thin film, as well as the measured nonlinear CD spectra. The inset shows a photographic image of the light spot of THG from the sample. (g) Power dependence of THG in logarithmic scale showing cubic power scaling law. The red dots show the measured data, the black dashed line is a fit to the data with a third-order power function.

Comment 1) Caption to figure 1 “tree typical modes” should presumably be “three typical modes”.

Response: Thank the reviewer for the careful reading. We corrected these typos in the revised manuscript.

Comment 2) This passage is misleading: “Although very recently q-BICs with versatile polarization responses (chirality selectivity) were theoretically proposed, they

required complicated 3D structures with broken symmetry in the propagation direction^{50,51,58} (Fig. 1b), hindering their implementation within current 2D patterning technology, especially at optical frequencies. Our present work shows that planar metasurfaces with only reduced in-plane symmetry can support q-BICs with both extrinsic and intrinsic planar chirality”. It implies that the new structure gives access to the same chiral optical response as these 3D structures just with simpler fabrication. This isn’t true and should be clarified.

Response: Thank the reviewer for this professional comment. Indeed, the previous proposed 3D structures host the 3D chiral properties, which are different from the 2D chiral properties supported by our proposed planar structures. The most remarkable difference between them is that, the CD for 3D chirality preserves the same polarization state between the input and output light, while the CD for 2D chirality is accompanied by circular conversion between input and output light due to the reciprocal requirement. In the revised manuscript, to avoid unnecessary misunderstanding, we clearly state that we are dealing with 2D chirality, please refer to “Our present work shows that planar metasurfaces with only reduced in-plane symmetry can support 2D extrinsic and intrinsic chirality with q-BICs” in line 17-22, page 3. We should also emphasize that, despite the difference between 3D and 2D chirality in terms of circular polarization conversion, many other properties in 3D chirality can indeed be achieved in 2D chirality system. As our newly added data about near-field enhancement CD and nonlinear CD indicate, the proposed DSS metasurface is not only capable of hosting perfect unitary CD in the far-field scattering, but also capable of supporting circular eigen-polarizations at Γ point, which is significant for active chiral manipulation of light. Here we add a more detailed and convincing discussion to explain how the designed planar chiral high-Q quasi-BIC metasurface is applied to chiral nonlinear enhancement. Please see Fig. 5 and supplementary Figs. S17-S18 and related text in the revised manuscript for details. It proves that the combination of polarization control and high-Q nonlinear enhancement is effective. Thus, as long as the structure supports a circular polarized eigen-polarization state, the nonlinear CD enhancement can be observed. So, the planar metasurface with the same DSS geometry can support either the near-unitary linear or

nonlinear CD.

Comment 3) It would be useful to check and include angles and values of the parameter δ smaller than 4 degrees and 20 nm to explicitly show the limits of Q for these structures.

Response: We thank Reviewer #1 for this constructive suggestion. Generally speaking, the Q factor increases with the decrease of the incident angle and the structure asymmetry δ . For convenience, we have supplemented the experiment with the incident angle of 2° as an interval as shown in the revised Fig. 3 (Fig. R2). It shows that, when the incident angles are $\pm 2^\circ$, we obtain the maximum Q factor of 602 with CD of 0.61. To further examine the limitation of Q-factor against structure asymmetry parameter δ , we have supplemented the simulation results for $\delta=2$ nm, 5 nm and 10 nm, respectively in Fig. S16 in the revised supplementary information (Fig. R3), the Q-factor and corresponding CD reach 800416/0.88, 66527/0.95, 12768/0.98, respectively. Due to the fabrication and measurement precision limitation, we do not experimentally fabricate metasurfaces with such small structure asymmetry. However, we believe that, the added simulation results already provide the guideline to understand the limits of Q for these structures. Please see Figs. 3c and 3d in the revised manuscript and Fig. S16 in the supplementary information for details.

Fig. R2. Experimental verification of the extrinsic planar chiral q-BIC with illumination symmetry breaking. (a) Experimental setup for the Jones matrix spectra measurement under the circular polarization basis. P_1 and P_2 represent the polarizers, QWP is the quarter-wave plate, L_1 , L_2 , and L_3 are lenses. (b) Schematic of the metasurface with $\delta=0$ supporting chiral q-BICs through different incident angles along the $\varphi=90^\circ$ direction. (c) Simulated and measured transmission Jones matrix spectra (T_{lr} , T_{rl} , T_{rr} and T_{ll}) for different incident angles ($\theta=0^\circ$, 2° , 4° , 6° , 8° , 10° , 12°) along the $\varphi=90^\circ$ direction. (d) Simulated and measured CD spectra for different incident angles extracted from the Jones matrix spectra in (c).

Fig. R3. Simulated transmission Jones matrix spectra of T_{ii} , T_{rr} , T_{rl} and T_{lr} as well as the CD spectrum of the metasurface with extremely small asymmetry parameters $\delta=2\text{nm}$, 5nm , and 10nm for normal incidences. The simulated Q-factor and corresponding CD reach $800416/0.88$, $66527/0.95$, $12768/0.98$, respectively.

Comment 4) I think it would be informative to plot the polarization eigenstate spectra for the simulated data for different parametric distances from the BIC. (This may also illuminate possible applications)

Response: We thank Reviewer #1 for this insightful suggestion. We have plotted the simulated polarization eigenstate spectra for different parametric distances (δ from 0 to 60 nm) from the BIC ($\delta=0$) (revised supplementary Fig. S10) (Fig. R4). We can see that the eigen-polarization state of the structure at the Γ point is always a linearly polarized eigenstate with the change of δ . In addition, the circular eigen-polarization state (non Γ point) will gradually move to the oblique incidence direction with the increase of δ .

Fig. R4. Eigen-polarization ellipse distribution in the k -space with different δ . The blue and red represent the eigen left-handed states and right-handed states. The red/blue dot represents the circular eigen-polarization state, and the black dot represents the V point (BIC).

Response to Reviewer #2

GENERAL COMMENTS

The authors demonstrate silicon metasurfaces sustaining high Q-factor modes that exhibit extrinsic and intrinsic chiral response. As the structure is planar, the CD is accompanied by circular polarization conversion and asymmetric transmission. The Q-factor of the metasurface is on the order of 10^2 . The results are technically sound, obtained with appropriate techniques, analyzed and interpreted very carefully, and presented in excellent detail. The conclusions are well supported by the results. However, chiral planar metasurfaces have been demonstrated in the literature (as also acknowledged by the authors e.g. refs. 14,15,24,38). Some of these featured high-Q resonances (e.g. Fano resonance in ref.14, q-BIC in ref.38). For these reasons I cannot recommend the publication of the submitted article in its present form to Nature Communications as it lacks of significant advancements compared to the established literature.

Response: Thank the reviewer for endorsing that our work is technically sound, obtained with appropriate techniques, very carefully analyzed and interpreted, excellently presented, and the conclusions are well supported by the results.

Our work differentiates from previous demonstrations of chiral planar metasurfaces as mentioned by the reviewer (cited in the revision as ref. 14, ref. 15 and ref. 38) in the following aspects.

First, to our best knowledge, all previous planar chiral metasurfaces could not achieve near-unitary and tunable chiroptical responses as well as controllable Q-factors as our work does with an assistant of the planar chiral q-BIC. Ref. 14, 15, and 24 were based on different physical mechanisms, namely, the Fano resonance or multi-mode interference of Mie-scatters. The typically achieved CDs and Q-factors were both relatively low ($CD < 0.7$, $Q < 100$), and the CD and Q-factor cannot be flexibly tuned by

one parameter in a regular way. In stark contrast, our present work uniquely demonstrates angle tunable CD and asymmetry parameter mediated inverse quadratic law of Q-factor based on the proposed planar chiral q-BIC metasurface with a DSS inclusion, which provides a clear guideline to flexibly design CD and Q-factor on demand.

Second, our work demonstrates new access to control both maximum linear and nonlinear CD assisted by planar q-BIC. Although ref. 38 represents another work by combining planar chiral metasurfaces with q-BIC, it only focuses on the nonlinear CD and is **solely based on numerical simulations, not experimentally verified yet**, while its linear CD does not reach the maximum. Instead, we experimentally demonstrated both the linear and nonlinear maximal CD [see Fig. 5, supplementary Figs. S17-S18 and related analysis in the revised version], which may lead to open directions such as sensitive asymmetry transmission and chiral emissions. Our present work proposes a DSS structure with sophisticated theoretical analysis and corroborated with solid experimental results, paving the way for practical applications. As the reviewer #1 endorsed, the importance of this paper is defined by the practical significance of high Q circular dichroism. Thus, we hope that our revised manuscript with clear innovations can meet the criterion to guarantee publications in Nature Communications.

A couple of additional points:

Comment 1) From the simulation results in Fig. 1e the BIC show a strong dispersion as the in-plane wave vector varies. What is the NA used in the measured transmittance spectra? Have the authors theoretically considered how a finite NA would affect the optical response?

Response: Indeed, as the reviewer #2 said, the dispersion of BIC is very strong. As we do not collect all in-plane wave vector components at one time, the finite NA has little effect on the optical response. To collect the data for a particular in-plane wave vector, we rotate the metasurface sample to obtain an oblique incidence setup with the demanded in-plane wave vector component. Actually, we only use a very low NA (0.1)

objective in our experiment. The strong dispersion here means that the quasi-BIC shifts rapidly to the long-wave band as the angle of incidence increases. This is not absolutely related to the NA of the collection objective. The area of the outgoing light will be the same as the area of the beam before the incident sample, and there will be no diffraction dispersion. In addition, the spectrometer needs to collect the transmitted light signal, so we only need the area of transmitted light to be less than the collection surface of the spectrometer. For normal incidence, it is easy to understand that the incident area only needs to be smaller than the collection surface of the spectrometer. In oblique incidences, we measured by rotating the sample, so the direction of incidence and transmission remains the same. The position of the collection surface remains unchanged, and the beam will not disperse. Therefore, it can also be measured without changing the NA of objective lens to focus on the spectrometer. We have also simulated this situation. As shown in Fig. R5, in the case of normal or oblique incidences, it only has transmitted, refracted and reflected in the beam direction without diffraction and scattering. Therefore, finite NA would not affect the optical response.

Fig. R5. The overall field patterns show the transmission in the same propagation direction of the incident light passing through the structure in the case of oblique

incidence.

Comment 2) The result in the abstract do not seem to match the main results of the paper. For the metasurface with Q-factor 390 it reads CD equal to 0.88.

Response: Indeed, the experimentally obtained highest Q-factor and the highest CD do not coincide for the same situation. To avoid misunderstanding, we revised the sentence and only claimed the maximal experimental $CD=0.93$ in the abstract. Although in simulations changing structure asymmetry (δ) will always maintain a high CD, it is understandable that the transmission peaks decrease at high Q-factors ($\delta=20$ nm). We have modified the description in the revised manuscript. Please see abstract in the revised manuscript for details.

Response to Reviewer #3

GENERAL COMMENTS

In this work, by means of numerical simulations the authors design a dielectric metasurface that exhibits a narrow resonance with high chirality. Also, the proposed system is experimentally verified, showing quantitatively and qualitatively agreement between simulations and measurements.

The design is based on the concept of bound states in the continuum (BICs) and asymmetric meta-atoms (without in-plane mirror symmetry). First, the eigenfrequencies of the proposed system is studied, showing a resonance with divergence Q-factor associated with the well-known symmetry-protected BIC at the Gamma point (in this system, related to the magnetic dipole response). Later, the circular dichroism (CD) is calculated using two different mechanisms to couple to the quasi-BIC are analyzed: (i) by changing the angle of incidence; (ii) by breaking the in-plane inversion symmetry of the meta-atoms (and working at normal incidence). In both cases narrow resonances with almost “ $CD = 1$ ” are found in the proximities of the BIC condition. Finally, the theoretical design is experimentally tested, validating the numerical results.

I personally think that the manuscript is solid, rigorous, well written and well presented. The good agreement between simulations and experiment makes the work

appealing. In addition, the results contribute to the progress in the current field. For these reasons I believe that the manuscript deserves publication in Nature Communications. Nonetheless, I have some minor concerns that would help to polish the manuscript.

Response: We are thankful to reviewer #3 for acknowledging that our manuscript is solid, rigorous, well written, well presented, contributes to the progress in the current field, and deserves publication in Nature Communications. We appreciate your effort in carefully reviewing our work. The responses to your valuable suggestions are appended below.

Comment 1) Ref. 58 is included in the sentence- “Although very recently q-BICs with versatile polarization responses (chirality selectivity) were theoretically proposed, they required complicated 3D structures with broken symmetry in the propagation direction”. In my understanding, in this work the proposed structure is not a “complicated 3D structure”, it is simply made of rods with slightly different heights and it can be considered a planar chiral metasurface. In order to improve the novelty statements of the current manuscript, I suggest introducing this reference in a different way.

Response: Thank Reviewer #3 for this valuable suggestion. In the revised manuscript, we removed the claim of simpler fabrication of our present design outperforming 3D chiral designs. And the introduction of Ref. 58 (Now it is Ref. 57) is revised to the sentence “Recently, chiroptical nanostructures mediated by BICs have been proposed, demonstrating perfect unitary chirality and extremely high Q-factors, and greatly expanding the available platforms to achieve optical chirality” in line 30-32, page 2.

Comment 2) In the first section of the manuscript all the results are obtained by means of numerical simulations, but this fact is not clearly said in the text. I encourage the author to include a sentence highlighting this point.

Response: Thank Reviewer #3 for this constructive suggestion. We added a sentence highlighting this point in the first section of the manuscript, please refer to “From the

simulated band structure and quality factor as shown in Figs 1(e and f), we see that the DSS metasurface hosts a BIC at the Γ point, characterized by a vertical magnetic dipole (MD) mode as shown in Figs. 1(e and f)” in line 24-25, page 3. “Figure 2b shows the calculated transmission spectra of all Jones matrix elements and the CD spectrum of the DSS metasurface with C_2 symmetry ($W_1=W_2$ and $L_1=L_2$, but $L_1 \neq W_1$) at oblique incidence ($\theta=8^\circ$, $\varphi=90^\circ$)” in line 12-14, page 4. “(e) Simulated band structure”, “(f) Simulated Q-factors” in line 8-11, page 13.

Comment 3) Again, in the conclusions it is said that

- “the intrinsic planar chirality can be achieved under normal incidence with maximum CD of 0.99 (theory) and 0.93 (experiment)”.

It should be read “simulation” instead of “theory”.

Response: We agree that the meaning of “theory” we express here is simulation. So, to address this point, we made the following modifications in the revised manuscript.

Comment 4) From the eigenmode analysis, the metasurface supports three resonances (at the frequency windows under study), but only the BIC resonance is observed in the CD maps of Fig. 2 c and g. Could the authors comment on this fact? I would also expect some (broad) features around the other resonant frequencies since the meta-atom is chiral per se.

Response: Indeed, the metasurface has three modes near the chiral BIC. The main reason we show is the comparison between the high-Q quasi-BIC mode and low-Q modes (Figs. 1e and 1f). It can be seen that the Q-factors of the other two modes are less than 10, and the calculated field enhancement is only about 10 (Fig. R6). Due to the low Q factor and low field enhancement, the light intensity is similar to that of the incident light, indicating that there is almost no resonance, just like the extended wave. Therefore, it cannot be observed in Figs. 2c and 2g. The simulation confirms that the mode intensity is too weak to be recognized in the spectra. There might be high-Q chiral characteristics near other distant wavelengths, but in the working band we designed here, this chiral BIC mode is dominant.

Fig. R6. Field enhancement of three modes. (a) Simulated transmission Jones matrix spectra of $T_{||}$, T_{rr} , T_{rl} and T_{lr} as well as the CD spectrum of the metasurface with $\delta=0$ under oblique incidence ($\theta=8^\circ$, $\varphi=90^\circ$). (b-d) Representative near-field $|H/H_0|^2$ plot for three different modes.

Comment 5) It looks like the experimental results shown in Fig.3 and 4 are noisier for low transmittance than for higher one. Then, the narrow resonances are well resolved, but one can think that noise is “numerically” removed. It would be valuable to include some comments about that.

Response: Thank reviewer #3 for this comment. Due to the signal-to-noise ratio of the spectrometer, when the signal is strong, the signal light is much larger than the noise light, and the measurement result is more accurate (the position of the peak); When the signal is weak, the signal light is at the same level of or lower than the noise light, so the measured results are mainly oscillatory noise signals. In addition, the measurement precision interval of the spectrometer in experiment is constant. If the measured spectral amplitude difference is large and the bandwidth is narrow, and the interval in the same wavelength range is

certain, the resonance with high Q and large amplitude difference must have fewer data points than the wide spectra with low transmittance. It can also be explained that it is not that the low transmittance noise is greater than the high transmittance, but that the low-Q resonance noise is greater than the high-Q resonance. The high-Q resonance of low noise here can be the peak of high transmittance or the dip of low transmittance. So, there is less noise. It is determined by the Q-factor of resonances.

Comment 6) It is interesting that in the comparison of simulations and experiments, the results of Fig. 3 (illumination symmetry breaking) shown similar resonance widths, while for Fig. 4 (broken meta-atom in-plane inversion symmetry) the experiment is broader than simulation. Can this be related to the structure tolerance as studied in Fig. S13?

Response: Thank Reviewer #3 for careful assessment. Indeed, this discrepancy is related to the structure tolerance of the fabricated structure. The structure in Fig. 3 ($\delta=0$) is the easiest one for experimental fabrication, because this structure is completely symmetrical and the radius of each side is relatively large. Under other parameters, especially for $\delta=80$ nm, the radius at the long edge is very small and the corner angle is very sharp. The fabrication precision demand for those parameters is more stringent, leading to the discrepancy of the bandwidth between the simulation and experimental results. Generally speaking, the larger the delta, the more difficult for precise fabrication and the less consistency between the simulation and experimental results. In addition, the change of incident angle is much easier than the change of the structure asymmetry δ , as we deal with the same structure, no structure tolerance is involved. To address this point, we have added related discussion in the revised supplementary manuscript, please refer to line 8-18, page 20.

Comment 7) In the “Author Contribution” section, “G. H. A. O” is written (without the comma that separates the names).

Response: Thank the reviewer for careful reading. We corrected these typos in the revised manuscript.

REVIEWERS' COMMENTS

Reviewer #1 (Remarks to the Author):

The authors have adequately addressed most of my comments and have also performed new nonlinear measurements showing enhanced nonlinear circular dichroism, which I find interesting and does point to potential applications. I therefore believe the paper is suitable for publication in Nature Communications.

However, I still believe that claiming that this work can be applied to "chiral sensing, enantiomer selection, and chiral quantum emitters" is misleading. The measurement of much greater THG for RCP pumping vs LCP pumping does demonstrate the chiral nature of the nanostructure, but the nearfield shown in Fig.S17 is clearly dominated by the out of plane component of the electric field and is not chiral. In other words, placing a chiral emitter/scatterer, whether a quantum dot or molecule, at the center of this structure may modify the farfield polarization emitted/scattered, but it would not selectively enhance the response of an emitter/scatterer with a particular handedness. Assuming the transmission for the structure in fig.5d is still dominated by conversion, i.e. t_{lr}, then reciprocity guarantees the same nearfield enhancement would be generated from pumping the device with LCP from the other side, which explains the lack of chirality in the nearfield. If this isn't the case, and this second mode the authors have introduced breaks the out of plane mirror symmetry in some way, this should also be discussed in more detail. As stated above, I still believe the structures introduced will have interesting uses for chiral lasers or nonlinear filters etc, but in my opinion the paper, in its current form, over states the range of applications.

Reviewer #2 (Remarks to the Author):

The authors have fully addressed all the reviewers' concerns. In particular they have provided a more careful and convincing discussion on the novelty and impact of their work. Moreover they added a new experiment on nonlinear CD at the third-harmonic. For these reasons I believe that the submitted manuscript deserves publication in Nature Communications.

Reviewer #3 (Remarks to the Author):

The authors have successfully addressed the referees comments and have followed the referees recommendations, improving the quality of the manuscript. Therefore, from my side I think that the manuscript is suitable for publication. Nonetheless, I agree with the opinion of the other referees about the lack of novelty of the first version of the manuscript. Since my concerns were "weaker" than theirs, I understand that their opinion will be more determinant for the final decision.

Response to Reviewer #1

The authors have adequately addressed most of my comments and have also performed new nonlinear measurements showing enhanced nonlinear circular dichroism, which I find interesting and does point to potential applications. I therefore believe the paper is suitable for publication in Nature Communications.

However, I still believe that claiming that this work can be applied to “chiral sensing, enantiomer selection, and chiral quantum emitters” is misleading. The measurement of much greater THG for RCP pumping vs LCP pumping does demonstrate the chiral nature of the nanostructure, but the nearfield shown in Fig.S17 is clearly dominated by the out of plane component of the electric field and is not chiral. In other words, placing a chiral emitter/scatterer, whether a quantum dot or molecule, at the center of this structure may modify the farfield polarization emitted/scattered, but it would not selectively enhance the response of an emitter/scatterer with a particular handedness. Assuming the transmission for the structure in fig.5d is still dominated but conversion, i.e. tlr, then reciprocity guarantees the same nearfield enhancement would be generated from pumping the device with LCP from the other side, which explains the lack of chirality in the nearfield. If this isn't the case, and this second mode the authors have introduced breaks the out of plane mirror symmetry in some way, this should also be discussed in more detail. As stated above, I still believe the structures introduced will have interesting uses for chiral lasers or nonlinear filters etc, but in my opinion the paper, in its current form, over states the range of applications.

Response: We thank Reviewer #1 for the professional comments. As the circular eigen-polarizations in our planar chiral metasurface are obtained at the far-field radiation region, the chirality is basically reflected in the far-field radiation or emission. The polarization character in the near-field is not considered at the current status. Therefore, we followed the reviewer's comment to change our potential applications of “chiral sensing, enantiomer selection, and chiral quantum emitters”, into “chiral lasers and nonlinear filters”. Please refer to the last sentence of the Abstract and Discussion of the revised manuscript and Line 12, Page 3.